# Molecular Phylogenetic Relationships and Unveiling Novel Genetic Diversity among Slow and Pygmy Lorises, including Resurrection of *Xanthonycticebus intermedius*

**DOI:** 10.3390/genes14030643

**Published:** 2023-03-03

**Authors:** Mary E. Blair, Giang T. H. Cao, Elora H. López-Nandam, Daniel A. Veronese-Paniagua, Mark G. Birchette, Marina Kenyon, Badrul M. Md-Zain, Rachel A. Munds, K. Anne-Isola Nekaris, Vincent Nijman, Christian Roos, Hoàng M. Thach, Eleanor J. Sterling, Minh D. Le

**Affiliations:** 1Center for Biodiversity and Conservation, American Museum of Natural History, New York, NY 10024, USA; elopez-nandam@calacademy.org (E.H.L.-N.); d.andre.veronese@gmail.com (D.A.V.-P.); mark.birchette@liu.edu (M.G.B.); sterling@amnh.org (E.J.S.); 2Department of Genetics, Vietnam National University, Hanoi 10000, Vietnam; gjag.ga.kon@gmail.com; 3Institute for Biodiversity and Sustainability Science, California Academy of Sciences, San Francisco, CA 94118, USA; 4The Division of Biology & Biomedical Sciences, Washington University in St. Louis, St. Louis, MO 63110, USA; 5Department of Biology, Long Island University Brooklyn, Brooklyn, NY 11201, USA; 6Dao Tien Endangered Primate Species Centre, Dong Nai 76000, Vietnam; marina@monkeyworld.org; 7Faculty of Science and Technology, Universiti Kebangsaan Malaysia, Bangi Selangor 43600, Malaysia; abgbadd1966@yahoo.com; 8Department of Anthropology & Archeology, University of Calgary, Calgary, AB T2N 1N4, Canada; rmunds27@gmail.com; 9Nocturnal Primate Research Group, Oxford Brookes University, Oxford OX3 0BP, UK; anekaris@brookes.ac.uk (K.A.-I.N.); vnijman@brookes.ac.uk (V.N.); 10School of Social Sciences and Centre for Functional Genomics, Oxford Brookes University, Oxford OX3 0BP, UK; 11Gene Bank of Primates and Primate Genetics Laboratory, German Primate Center, Leibniz Institute for Primate Research, 37077 Göttingen, Germany; croos@dpz.eu; 12Department of Anthropology, Vietnam National University, Hanoi 10000, Vietnam; thachmaihoang1979@yahoo.com; 13Department of Geography & Human Ecology, Rutgers, The State University of New Jersey, New Brunswick, NJ 08854, USA; 14Faculty of Environmental Sciences, University of Science and Central Institute for Natural Resources and Environmental Studies, Vietnam National University, Hanoi 10000, Vietnam

**Keywords:** historical DNA, Lorisidae, mitochondrial DNA, molecular genetics, museum specimens, strepsirrhines

## Abstract

Genetic analysis of historical museum collections presents an opportunity to clarify the evolutionary history of understudied primate groups, improve taxonomic inferences, and inform conservation efforts. Among the most understudied primate groups, slow and pygmy lorises (genera *Nycticebus* and *Xanthonycticebus*) are nocturnal strepsirrhines found in South and Southeast Asia. Previous molecular studies have supported five species, but studies using morphological data suggest the existence of at least nine species. We sequenced four mitochondrial loci, *CO1, cytb, d-loop*, and *ND4*, for a total of 3324 aligned characters per sample from 41 historical museum specimens for the most comprehensive geographic coverage to date for these genera. We then combined these sequences with a larger dataset composed of samples collected in Vietnam as well as previously published sequences (total sample size N = 62). We inferred phylogenetic relationships using Bayesian inference and maximum likelihood methods based on data from each locus and on concatenated sequences. We also inferred divergence dates for the most recent common ancestors of major lineages using a BEAST analysis. Consistent with previous studies, we found support for *Xanthonycticebus pygmaeus* as a basal taxon to the others in the group. We also confirmed the separation between lineages of *X. pygmaeus* from northern Vietnam/Laos/China and southern Vietnam/Cambodia and included a taxonomic revision recognizing a second taxon of pygmy loris, *X. intermedius*. Our results found support for multiple reciprocally monophyletic taxa within Borneo and possibly Java. The study will help inform conservation management of these trade-targeted animals as part of a genetic reference database for determining the taxonomic unit and provenance of slow and pygmy lorises confiscated from illegal wildlife trade activities.

## 1. Introduction

Molecular research on historical museum collections [1] presents a remarkable opportunity to clarify the evolutionary history of understudied or undersampled primate groups [2,3]. For example, although primates from the infraorder Lorisiformes (galagids and lorisids) represent nearly 10% of living primate species, their taxonomy, evolutionary history, and biogeography remain poorly understood [4]. This is likely due to the nocturnal habits of these taxa as well as the highly cryptic morphologies and low densities associated with nocturnality, all of which have likely impeded research progress to uncover species diversity in this group [4]. Lorisiforms are, therefore, ideal candidates to leverage the sampling of museum collections to improve our understanding of their evolutionary history.

Especially for globally threatened primates, an assessment of genetic relationships within understudied groups can improve taxonomic inferences and inform conservation efforts [5,6,7]. While most lorisiforms are threatened with extinction, the slow lorises, of the genus *Nycticebus* É. Geoffroy, 1812 and the pygmy loris of the genus *Xanthonycticebus* Nekaris and Nijman, 2022 within the Asian lorisids (subfamily Lorisinae), are acutely imperiled by the rampant demand for wildlife trade [8,9,10,11]. Resolving slow and pygmy loris taxonomy is thus urgently needed to inform trade management and mitigation strategies as well as rehabilitation and reintroduction efforts [12,13,14].

Within the slow and pygmy lorises, there are currently nine species recognized across South and Southeast Asia [4,7,15,16,17,18,19,20], including the greater slow loris (*Nycticebus coucang*), the Bengal slow loris (*Nycticebus bengalensis*), the Javan slow loris (*Nycticebus javanicus*), the Bornean slow loris (*Nycticebus borneanus*), the Philippine slow loris (*Nycticebus menagensis*), the Kayan slow loris (*Nycticebus kayan*), the Sumatran slow loris (*Nycticebus hilleri*), and Sody’s slow loris (*Nycticebus bancanus*). A new genus name was recently proposed for the ninth species, the pygmy loris (*Xanthonycticebus pygmaeus*; [18]). Most molecular analyses to date consistently support five of the species, *X. pygmaeus, N. coucang, N. menagensis, N. javanicus and N. bengalensis*, as reciprocally monophyletic (e.g., [17,21]), but the geographic sampling of molecular studies within each of these taxa has not been sufficient to date to validate the morphologically based findings of Munds et al. [16] demonstrating the distinction of *N. bancanus, N. borneanus* and *N. kayan*, or Nekaris and Jaffe [20]’s support for *N. hilleri.*

A comprehensive evaluation of the molecular relationships within the slow and pygmy lorises has been challenging to date because broad geographic sampling is not available for most species. Further, as summarized by Masters [22], while hidden genetic and morphological diversity within *Nycticebus* has been recognized for many years, opinions differ over whether that diversity is at the species or subspecies level. In addition to challenges of geographic sampling, pelage coloration changes during ontogeny, and for at least pygmy lorises and Javan slow lorises, seasonally, make it difficult to distinguish between morphological variations and diversity. However, additional species-level lineages within the group are likely. In addition to the species on Borneo and Sumatra supported by the morphological research of Munds et al. [16] and Nekaris and Jaffe [20], there is also potential for undescribed taxa in mainland Southeast Asia within the distributions of *N. bengalensis* and *X. pygmaeus*. Both species have widespread distributions, and studies to date have not included samples from across their full ranges. Further, there is considerable pelage variation in both species; for the Bengal slow loris, pelage variation appears to follow geography [23], but for the pygmy loris, this variation is likely seasonal [24,25]. For many other species complexes of primates and other vertebrates with similar distributions to the pygmy loris (east of the Mekong), there is a clear latitudinal turnover of endemic diversity (doucs, gibbons, and other mammals; [26,27,28]).

In order to test existing hypotheses for phylogenetic relationships and illuminate any novel diversity, we sampled historical museum specimens and conducted fieldwork to amass the most geographically comprehensive molecular dataset for slow and pygmy lorises to date. We sampled extensively from museum collections in both the U.S. and Vietnam, with the aim to cover the full taxonomic and geographic range of both *Nycticebus* and *Xanthonycticebus*. We also conducted fieldwork in Vietnam to gather more focused sampling on the putative zoogeographic barrier for Mainland Southeast Asian taxa. From this dataset, we sequenced four mitochondrial loci and combined these sequences with previously published information and inferred phylogenetic relationships using Bayesian inference and maximum likelihood methods. To further understand slow loris evolutionary history and biogeography, we used the results of our phylogenetic analysis to infer divergence dates for the most recent common ancestors of major lineages, which should be considered preliminary pending analysis of nuclear DNA sequences.

## 2. Materials and Methods

### 2.1. Sample Collection

Following approved protocols for destructive sampling as established by each respective institution (e.g., see https://www.amnh.org/research/vertebrate-zoology/mammalogy/collection-information/destructive-sampling (accessed on 10 February 2023)), we sampled historical specimens of slow and pygmy lorises from across all putative taxa and geographic ranges for a total of 71 historical samples that were collected between 1884 and 1980; samples were collected in the U.S. from the American Museum of Natural History (AMNH) Mammalogy collection, the Smithsonian National Museum of Natural History (NMNH), the Field Museum of Natural History (FMNH), and in Vietnam from the Hanoi Zoological Museum, the Da Lat Biological Museum, and the Institute of Ecology and Biological Resources’ (IEBR) Zoological Museum. Details of the samples included in this study from each institution are summarized in Table 1 and Appendix A, and geographic locations are shown in Figure 1. Samples included 2 × 2 × 3 mm size samples taken from skins (mostly finger or toepads) or dried connective tissue remaining on skeletons. For each museum specimen, photographs and measurements (Appendix A) were taken following standard species identification protocols for slow and pygmy lorises as used in previous studies [16,23] using digital calipers accurate to 0.1 mm and with a color card standard (Calibrite). External measurements (head-body length, tail length, hind foot length, and ear length) were also recorded when available on original museum specimen labels.

We analyzed cranial measurements for adult specimens of pygmy lorises using standard descriptive statistics, unpaired *t*-tests, and a principal components analysis (PCA; in R version 4.2.1) to compare nine linear measures (greatest skull length, mandible length, palate length, palate breadth, biorbital breadth, staphylion to basion length, bicanine breadth, ramus height, bizygomatic breadth; for definitions, see Appendix A) among identified lineages and specimen sex. For the PCAs, all measurement values were standardized by subtracting the mean and dividing by the standard deviation. Measurements of three additional specimens with known locality information were included from the Muséum National d’Histoire Naturelle (MNHN) but were not sampled for genetic analysis (Appendix A and Appendix A).

To improve geographic sampling for the putative zoogeographic barrier in Vietnam, we collected additional samples opportunistically during field surveys that were conducted across Vietnam from 2013 to 2015 as a part of a larger study to update the conservation status of slow and pygmy lorises in the country (Figure 1; see Table 1 and Appendix A for a list of all sites). At each surveyed site, we followed established methodologies for surveying nocturnal mammals, specifically reconnaissance survey techniques [29] for spotlight surveys, following the methods of other published slow loris surveys (e.g., [30,31]). Trails, roads, dry riverbeds and cut transects were walked slowly (0.5–1 km/hr) by a team of 2–3 people spaced at least 10 m apart. Surveys began at dusk and lasted 2–5 hrs. Observers scanned all levels of vegetation using headlamps with red filters to detect loris’ distinctive orange eyeshine. After spotting potential eye shine, a halogen spotlight was used to confirm species identification with binoculars. When possible, photographs were taken of sighted lorises, opportunistic fecal samples were collected, or hair samples were collected after hand capture for genetic analysis. We stored fecal samples in 8 mL plastic tubes with RNAlater buffer (Ambion, Austin, TX, USA) at 4 °C in the field and −20 °C in the laboratory, and hair samples were kept dry in a sealed plastic bag. This research was performed with appropriate research and sample collection approval from Vietnam’s Forestry Administration, and the animal care protocol for this research project was approved on 4 June 2014 by the American Museum of Natural History’s Institutional Animal Care and Use Committee (IACUC).

In total, 51 new samples were collected, extracted, and sequenced for this study, as described below, in addition to the samples from museum specimens. An additional 13 already sequenced individuals by co-authors, collaborators from previously published studies [21,32], and one unpublished mitogenome, including seven samples mined from GenBank and one unpublished mitogenome (aDNA716, see Table 1) that was generated following the methods described in Roos et al. [33]. These additional individuals were included in the study to achieve our desired level of taxonomic and geographic coverage and as outgroups (see Table 1 and Appendix A for full details and references for each included sequence).

### 2.2. Molecular Data Collection

All DNA from USA museum specimens were extracted in the AMNH ICG Ancient Facility using a Qiagen DNeasy Blood and Tissue Kit (Qiagen, Germany) with modifications to the manufacturer’s instructions for animal tissue to increase the amount of DNA obtained (Qiagen). After adding buffer ATL and proteinase K, we placed the samples in a heat shaker at 56 °C for 18–24 h. We then added another 180 uL and 20 uL of buffer ATL and proteinase K, respectively, and the samples were incubated at 56 °C for an additional 18–24 h until complete digestion. Buffer AL and ethanol quantities were thus doubled to stop the digestion reaction. Additionally, before the addition of buffer AE, we heated it to 70 °C and incubated the spin columns for 30–40 min before the final elution step. All other steps in the Tissue Handbook Protocol were followed as indicated. A negative control was used for every 6 samples extracted. The Ancient Facility is a separate workspace dedicated to historical tissues that include independent sets of reagents in order to reduce the risk of contamination from exogenous DNA and PCR-amplified products.

Field- or confiscation-collected samples and samples from Vietnamese museums were extracted at the genetic laboratory at the University of Science, Vietnam National University in Hanoi. Museum samples treated with formalin were extracted using GTE and Puregene Tissue Kit (Qiagen, Germany) following the manufacturer’s instructions to recover even small amounts of residual undegraded DNA [34]. Other samples were processed using the same protocol used at the AMNH ICG.

While methods to collect genomic data from museum specimens are available, these require quality standards that less than 1/4 of our samples met in terms of data concentration and quality. To achieve the largest and most complete sample coverage in terms of what was available with current technologies, we used Sanger sequencing of mtDNA.

All generated DNA sequence data have been deposited to GenBank under the following accession numbers: OQ518052-OQ518145, OQ555473-OQ555600 (Table 1 and Appendix A). From the new samples described above, we amplified four mitochondrial genes: *cytochrome b* (*cytb*) (1140 bp), *NADH dehydrogenase subunit 4* (*ND4*) (716 bp), *cytochrome c oxidase subunit 1* (*CO1*) (927 bp), and tRNAs *(-Thr* and -*Pro*) + *d-loop* (541 bp), using primer pairs listed in Appendix A for a total of 3324 bp per sample. The *cytb* sequence includes the full *cytb* gene as well as approximately 35 bp of *tRNA-Glu* (from position 14,765 to 15,386 on the mitogenome). The *ND4* sequence includes the first 686bp of *ND4* as well as 30 bp of *ND4L* (from position 10,190 to 10,905 on the mitogenome). The *CO1* sequence includes the first 856 bp of *CO1* as well as the full *tRNA-Tyr* (67 bp) and 13 bp of *tRNA-Cys* (from position 5286 to 6246 on the mitogenome). The tRNAs + *d-loop* sequence includes the first 344 bp of the *d-loop* as well as the full *tRNA-Pro* (69 bp) and *tRNA-Thr* (72 bp) (from position 15,387 to 15828 on the mitogenome). These loci were chosen because they are known for containing an appropriate level of variation among species in most vertebrates [35,36,37].

Since commonly only highly degraded DNA can be extracted from museum specimens, we used several sets of primers for historical samples (Appendix A) to target overlapping segments of the target mitochondrial genes. Each 26 μL PCR reaction consisted of 6.0 μL water, 12.5 μL Top Taq Master Mix, 1 μL bovine serum albumin (BSA), 1 μL each of 10 μM forward and reverse primer, 2.5 μL CoralLoad Concentrate-10× (Qiagen), and 2.0 μL template. Rohland and Hofreiter [38] found that the addition of BSA increases the success rate of extraction from historical specimens by overcoming inhibitors in the samples.

PCR amplification was performed with a touchup program with the following conditions: initial denaturation at 94 °C for two minutes; five cycles, each consisting of 95 °C for 30 s, 51 °C for 30 s, and 72 °C for 90 s; 35 cycles, each consisting of 95 °C for 30 s, 55 °C for 30 s (annealing temperature varied for some primer pairs, see Appendix A), and 72 °C for 90 s; and a final extension stage at 72 °C for five minutes.

Before sequencing, we purified the amplification products of excess nucleotides, primers, enzymes, and other leftover PCR reagents using the AMPure protocol (AgentCourt Bioscience). We then cycle-sequenced purified PCR products with ABI BigDye Terminator Ready Reactions kits and electrophoresed our samples on an ABI 3730 DNA Analysis System. PCR products generated at Vietnam National University were sent to Firstbase (Malaysia) for sequencing. We carried out base calling using Sequencing Analysis (ABI) and assembled sequences using Geneious (Biomatters).

### 2.3. Phylogenetic Analysis

Sequences generated in this study were edited and checked for quality with 4-6x coverage using Geneious (Biomatters) and then aligned with previously published sequences for a total sample size of N = 153, including 71 historical specimens. We then trimmed the dataset to remove identical sequences as well as samples with major data gaps for a final dataset of N = 62, including 41 historical specimens (Table 1). Homologous sequences from the complete mitochondrial genomes of the red slender loris *Loris tardigradus* (NC_012763) and the southern lesser galago *Galago moholi* (KC757396) available on GenBank were included as outgroups. The final alignment length after trimming for gaps was 3324 aligned characters.

We inferred phylogenetic relationships using Bayesian inference (BI) as implemented in MrBayes v3.2.7 [39] and maximum likelihood (ML) in IQ-TREE v1.6.8 [40]. For BI, we performed both single and multiple models by codon partitions to examine the robustness of the tree topology [41,42]. Analyses were conducted with a random starting tree and run for 1 × 10^7^ generations with four Markov chains (one cold, three heated) with default settings. Values of sample points were plotted against the number of generations to detect the stationarity of the Markov chains. Trees generated prior to stationarity were removed from the final analyses using the burn-in function. Two independent analyses were performed simultaneously. The cut-off point for the burn-in function was set to 58 and 63 in the single- and multiple-model Bayesian analyses, respectively, as -lnL scores reached stationarity after 58,000 and 62,000 generations in both runs of the two analyses. The posterior probability (PP) values for all clades in the final majority rule consensus tree were provided.

For ML and BI analyses, we used the optimal model calculated by jModelTest v.2.1.10 [43]. The optimal model for nucleotide evolution was set to TVM+G for single-model BI and ML analyses. For the Bayesian multiple model analysis, nine models, TVM+G+X, HKY+I+X, TIM+I+X, GTR+I+G+X, GTR+I+X, SYM+I+G, TRN+I+G+X, HKY+G+X, and TRN+I+X, selected by PartitionFinder v2 [44] were assigned to ten partitions in MrBayes using the command APPLYTO. Model parameters were inferred independently for each data partition using the UNLINK command. Nodal support was also evaluated by PP in MrBayes and ultrafast bootstrap (UFB) (10,000 replications) in IQ-TREE. We regarded UFB values of ≥95% and PP values ≥ 0.95 as strong support for the monophyly of a given clade [39,45].

We also inferred divergence dates for the most recent common ancestors (MRCA) of major lineages using a BEAST analysis. We selected the relaxed-clock method [46] to estimate divergence times. The obtained dataset was used as input for the computer package BEAST v1.10.1 [47]. A priori criteria for the analysis were set in the program BEAUti v1.10.1. One calibration point, the origin of Asian lorisids estimated at 29.3 ± 6.2 million years ago (mya) [48], was used to calibrate the phylogeny. A general time-reversible (GTR) model using γ + invariant sites with four γ categories was employed, along with the assumption of a relaxed molecular clock. As for the priors, we used all default settings except for the Tree Prior category, which was set to incomplete birth death, as recommended for lineage-level analyses. The combined dataset was used for a single run. In addition, a random tree was employed as a starting tree. The length chain was set to 10^7^, and the Markov chain was sampled every 1000 generations. After the dataset was analyzed in BEAST, the resulting likelihood profile was then examined with the program Tracer v1.6 to determine the burn-in cut-off point. The final tree with calibration estimates was computed using the program TreeAnnotator v1.10.1. as recommended in the BEAST program manual.

**Table 1 genes-14-00643-t001:** Sample details, including GenBank accession numbers of mitochondrial sequences generated by or used in this study.

Species Name	English Name	Specimen No.	Map ID	Provider	Provenance	Clade	Reference	Genbank Acc. No.
*N. coucang*	Sunda slow loris	AMNH20870	26	AMNH	Zoo origin	1-m	This study	OQ518055 OQ555566
*N. coucang*	Sunda slow loris	FMNH98478	131	FMNH	Malaysia, Bentong	1-m	This study	OQ518069
*N. coucang*	Sunda slow loris	NMNH283915	116	NMNH	Malaysia	1-m	This study	OQ518063 OQ555488
*N. coucang*	Sunda slow loris	NMNH300000	66	NMNH	Malaysia, Selangor	1-m	This study	OQ518056 OQ555477
*N. coucang*	Sunda slow loris	NMNH355347	109	NMNH	Malaysia, Selangor	1-m	This study	OQ518058
*N. coucang insularis*	Sunda slow loris	NC040292	NC	J. Rovie- Ryan	Malaysia, Tioman Island	1-m	[49]	NC040292
*N. coucang malayanus*	Sunda slow loris	NMNH488075	112	NMNH	Malaysia, Pahang	1-m	This study	OQ518062 OQ555486
*N. coucang malayanus*	Sunda slow loris	NMNH488076	120	NMNH	Malaysia, Perak	1-m	This study	OQ518066 OQ555489
*N. coucang malayanus*	Sunda slow loris	NMNH488078	119	NMNH	Malaysia, Selangor	1-m	This study	OQ518065 OQ555478
*N. coucang malayanus*	Sunda slow loris	NMNH488079	122	NMNH	Malaysia, Selangor	1-m	This study	OQ518068 OQ555491
*N. bengalensis*	Bengal slow loris	aDNA716	716	NHMUK	India, Kohima, Naga Hills, Assam	1-nb	This study	OQ518145 OQ555539 OQ555565
*N. bengalensis+*	Bengal slow loris	AJ309867	AJ	Unknown	Unknown	1-nb	[50]	AJ309867
*N. bengalensis*	Bengal slow loris	AMNH112990	30	AMNH	N. Burma, Chindwin River below Haibum, Sagaing	1-nb	This study	OQ518075 OQ555497 OQ555542
*N. bengalensis*	Bengal slow loris	AMNH112991	29	AMNH	N. Burma, Singkaling Hkamti, Sagaing	1-nb	This study	OQ518074 OQ555496 OQ555541
*N. bengalensis*	Bengal slow loris	KC977312	KC9	H. Somura	Japan, Nasu World Monkey Park	1-nb	[51]	KC977312
*N. bengalensis*	Bengal slow loris	L21	L21	M. Le	Vietnam, Sa Pa	1-nb	This study	OQ518071 OQ555519 OQ555560
*N. bengalensis*	Bengal slow loris	L34	L34	M. Blair	Vietnam, Phú Quốc Island	1-nb	This study	OQ518073 OQ555532 OQ555561
*N. bengalensis*	Bengal slow loris	NMNH300015	99	NMNH	Thailand	1-nb	This study	OQ518077 OQ555482 OQ555550
*N. bengalensis*	Bengal slow loris	AMNH240010	27	AMNH	Thailand, Chanthaburi, E Base Khao Sai Dao Tai	1-tb	This study	OQ518085 OQ555540
*N. bengalensis*	Bengal slow loris	KC757405	KC7	H. Schulze	Unknown (zoo origin)	1-tb	[52]	KC757405
*N. bengalensis*	Bengal slow loris	NMNH296513	93	NMNH	Thailand, Kanchanaburi	1-tb	This study	OQ518086
*N. bengalensis*	Bengal slow loris	NMNH355064	102	NMNH	Thailand, Chiang Mai	1-tb	This study	OQ518088 OQ555484 OQ555552
*N. bengalensis*	Bengal slow loris	NMNH535153	101	NMNH	Thailand, Ubon Ratchathani	1-tb	This study	OQ518087 OQ555483 OQ555551
*N. hilleri*	Sumatran slow loris	NMNH267400	114	NMNH	Sumatra, Siantar	1-h	This study	OQ518054 OQ555487 OQ555554
*N. hilleri*	Sumatran slow loris	NMNH270595	113	NMNH	Sumatra, Siantar	1-h	This study	OQ518053 OQ555553
*N. bengalensis*	Bengal slow loris	LANGKAWI	LK	B. Md Zain	Malaysia, Langkawi Island	1	[32]	OQ466050
*N. bengalensis*	Bengal slow loris	NMNH114151	108	NMNH	Malaysia, Johor Lama	1	This study	OQ518078
*N. bengalensis*	Bengal slow loris	NMNH258870	91	NMNH	E Thailand, Chanthaburi	1	This study	OQ518076 OQ555481 OQ555549
*N. bengalensis*	Bengal slow loris	FMNH99616	124	FMNH	Thailand, Tak, Mae Sod, Ban Mae Lamao	1	This study	OQ518079 OQ555492 OQ555555
*N. coucang tenasserimensis*	Sunda slow loris	NMNH84389	105	NMNH	Thailand	1	This study	OQ518052
*N. coucang*	Sunda slow loris	NMNH14290	95	NMNH	Unknown	n	This study	OQ555476
*N. javanicus*	Javan slow loris	NJAV	NJ	H. Schulze	Java	2	[21]	KP410612
*N. javanicus*	Javan slow loris	N3	N3	H. Schulze	Java	2	[21]	KP410601
*N. kayan*	Kayan slow loris	AMNH106013	59	AMNH	Borneo, Peleben, Timur	3	This study	OQ518097 OQ555543
*N. kayan*	Kayan slow loris	FMNH108856	133	FMNH	Borneo, Sabah, Mt. Kinabalu	3	This study	OQ518099 OQ555494
*N. kayan*	Kayan slow loris	NMNH292553	87	NMNH	Borneo, Sabah, Mt. Kinabalu	3	This study	OQ518100 OQ555547
*N. kayan*	Kayan slow loris	NMNH317188	70	NMNH	Borneo, Sabah, Ranau	3	This study	OQ518098 OQ555479 OQ555546
*N. menagensis*	Philippine slow loris	FMNH129502	134	FMNH	Philippines, Tawi Tawi	3	This study	OQ518102
*N. menagensis*	Philippine slow loris	FMNH85926	128	FMNH	Borneo, Sabah, Tawau	3	This study	OQ518101 OQ555493 OQ555556
*N. bancanus*	Sody’s slow loris	NMNH142237	82	NMNH	Borneo, W Kalimantan, Sanggau	4	This study	OQ518092 OQ555480
*N. borneanus*	Bornean slow loris	AMNH17133	61	AMNH	Zoo origin	4	This study	OQ518090 OQ555545
*N. borneanus*	Bornean slow loris	NMNH142232	83	NMNH	Borneo, W Kalimantan, Landak River	4	This study	OQ518093
*N. kayan*	Kayan slow loris	AMNH16616	34	AMNH	Likely zoo or trade origin, tag notes ‘Singapore’	4	This study	OQ518094
*N. kayan*	Kayan slow loris	NMNH142239	79	NMNH	Borneo, W Kalimantan, Sanggau	4	This study	OQ518095
*N. menagensis*	Phillippine slow loris	AMNH32649	63	AMNH	Borneo, Ft. Kapit	4	This study	OQ518091
*N. kayan*	Kayan slow loris	AMNH106012	60	AMNH	Borneo, Peleben, Timur, Sungai Tajan	5	This study	OQ518089 OQ555544
*N. menagensis*	Phillippine slow loris	NMNH198857	77	NMNH	Borneo, E Kalimantan, Samarinda	5	This study	OQ518096
*N. javanicus*	Javan slow loris	AMNH101509	41	AMNH	Java, Soemedang	6	This study	OQ518084
*N. javanicus*	Javan slow loris	AMNH102027	5	AMNH	Java, Tasikmalaja, Tjamis	6	This study	OQ518080 OQ555567
*N. javanicus*	Javan slow loris	AMNH102845	40	AMNH	Java, Cheribon	6	This study	OQ518081
*N. javanicus*	Javan slow loris	NMNH521836	88	NMNH	Java, Tjilegong	6	This study	OQ518083 OQ555474 OQ555548
*N. coucang*	Sunda slow loris	NCOU751	751	Singapore Zoo	Malaysian Peninsula	7	[21]	KP410655
*N. coucang*	Sunda slow loris	ZSM3	ZS	ZSM	Sumatra, Batang Kwis	7	[21]	KP410591
*X. intermedius*	Northern pygmy loris	L0	L0	Huấn	Vietnam, Na Hang Nature Reserve, Tuyên Quang	8	This study	OQ518127 OQ555498 OQ555557
*X. intermedius*	Northern pygmy loris	KX397281	KX	Q. Ni	China, Geiju Zoo	8	[53]	KX397281
*X. intermedius*	Northern pygmy loris	L41	L41	M. Blair	Vietnam, Mương Lọ, Nam Đồng, Thừa Thiên Huế	8	This study	OQ518142 OQ555536 OQ555562
*X. intermedius*	Northern pygmy loris	L42	L42	M. Blair	Vietnam, Nam Đồng, Thừa Thiên Huế	8	This study	OQ518143 OQ555537 OQ555563
*X. intermedius*	Northern pygmy loris	HZM1001	N24	Vũ Ngọc Thành	Vietnam, Nam Giang, Quảng Nam	8	This study	OQ518144 OQ555538
*X. pygmaeus*	Southern pygmy loris	HZM1000	N23	Vũ Ngọc Thành	Vietnam, Chaval, Nam Giang, Quảng Nam	9	This study	OQ555564
*X. pygmaeus*	Southern pygmy loris	L2	L2	M. Blair	Vietnam, Lâm Đồng	9	This study	OQ518109 OQ555500 OQ555558
*X. pygmaeus*	Southern pygmy loris	L6	L6	M. Blair	Vietnam, Lâm Đồng	9	This study	OQ518113 OQ555504 OQ555559
*X. pygmaeus*	Southern pygmy loris	L40	L40	M. Blair	Vietnam, Thương Nhạt, Nam Đồng, Thừa Thiên Huế	9	This study	OQ555535
*L. tardigradus*	Red slender loris	NC012763	-			Outgroup	[54]	NC012763
*G. moholi*	Mohol bushbaby	KC757396	-			Outgroup	[52]	KC757396

+Mislabeled in GenBank as *N. coucang*. Species Name corresponds to corrected identifications from specimen tags based on labeled provenance, clade, and pelage-based identifications following Munds et al. [16]. Map ID represents how the sample is displayed in Figure 1. Clade number and color refer to the clades shown in Figure 2. Abbreviations: AMNH: American Museum of Natural History, New York, USA; FMNH: Field Museum of Natural History, Chicago, USA; HZM: Hanoi Zoological Museum, Hanoi, Vietnam; NMNH: National Museum of Natural History, Washington, D.C., USA; NHMUK: the Natural History Museum, London, UK; ZSM: Zoological State Collection, Munich, Germany.

## 3. Results

### 3.1. Inference of Phylogenetic Relationships

Phylogenetic analyses supported similar topologies (Figure 2). As expected, we found strong support for the basal position of *X. pygmaeus* as the sister taxon of all other slow loris species. Further, we found two reciprocally monophyletic lineages within *X. pygmaeus*, one composed of samples from northern Vietnam/Laos/China and one composed of samples from southern Vietnam/Cambodia (Figure 1 and Figure 2). The northern *X. pygmaeus* group is highly supported by both ML (UFB = 98%) and BI analyses (PP_partitioned_/PP_concatenated_ = 1.0/0.99), as is the southern *X. pygmaeus* group (UFB = 99%, PP_p_/PP_c_ = 1.0/0.99). It is noted that each group contains a sample from Quang Nam Province, Vietnam, but from different parts of the province. In terms of average pairwise genetic distance, the two clades differ by approximately 1.9% based on the combined mitochondrial dataset (Table 2 and Appendix A).

After *X. pygmaeus*, a weakly supported group that includes some *N. javanicus* and Sumatran *N. coucang* samples (PP_p_/PP_c_ = 0.64/0.71) branches off as sister to the rest of the genus *Nycticebus*. Within this group, the *N. javanicus* and Sumatran *N. coucang* samples are supported as reciprocally monophyletic to each other (UFB = 94%, PP_p_/PP_c_ = 0.60/0.57; UFB = 96%, PP_p_/PP_c_ = 0/85/0.69, respectively). Other *N. coucang* and *N. javanicus* specimens without a clear provenance (NMNH14290) or that we know originated in trade (NJAVA, N3) clustered elsewhere at the base of the *N. bengalensis* clade (Figure 2).

The samples from Borneo are supported as a monophyletic group within the genus (UFB = 100%, PP_p_/PP_c_ = 1.0/0.71), as are at least three distinct lineages within Borneo, namely a northern Borneo group, a western Borneo group, and an eastern Borneo group (each with strong support at UFB = 94%, PP_p_/PP_c_ = 0.92/0.53; UFB = 99%, PP_p_/PP_c_ = 0.87/0.88; UFB = 100%, PP_p_/PP_c_ = 0.96/0.96, respectively). The northern Borneo group also includes the sample described as *N. menagensis* from Tawi Tawi, and there is significant structuring within the western group. The western and eastern groups cluster together and are sisters to the northern group (UFB = 91%, PP_p_/PP_c_ = 0.62/0.73). However, samples that we identified phenotypically as *N. menagensis, N. kayan, N. bancanus*, and *N. borneanus* are assorted among these groups.

Samples of *N. coucang* from peninsular Malaysia are paraphyletic to the Sumatran *N. coucang* samples and instead cluster within the *N. bengalensis* group. The *N. coucang* samples from peninsular Malaysia are mostly comprised of the putative subspecies *N. c. malayanus* and are supported as distinct from the rest of the *N. bengalensis* group (UFB = 97%, PP_p_/PP_c_ = 0.87/0.82), with the putative *N. c. tennaserimensis* as a sister to this group and also distinct (UFB = 98%, PP_p_/PP_c_ = 1.0/0.99). The two Sumatran *N. hilleri* samples are supported as distinct (UFB = 99%, PP_p_/PP_c_ = 1.0/0.99) but only weakly supported as sisters to this group (UFB < 50%, PP_p_/PP_c_ = 0.52/0.74).

The rest of the *N. bengalensis* group is less strongly supported than other lineages as reciprocally monophyletic from the peninsular Malaysian *N. coucang* (UFB < 50%, PP_p_/PP_c_ = 0.66/<0.50) for the presented topology, in which the sample from Langkawi Island is included in this group. However, the group also includes some well-supported distinct lineages within it, including the putative ‘dark morph’ *N. bengalensis* group from southern Thailand (UFB = 99%, PP_p_/PP_c_ = 1.0/0.99) and a ‘northern’ *N. bengalensis* group that includes the sample from Phu Quoc Island (UFB = 59%, PP_p_/PP_c_ = 0.99/0.94).

### 3.2. Divergence Dating

We estimated 11.34 mya (95% highest probability density (HPD): 14.17–8.59 mya) for the *Xanthonycticebus* and *Nycticebus* MRCA (Figure 3). We also estimated a wide gap (roughly 6 million years) between the divergence of *N. coucang* (Sumatra) and the Java 2 clade and that of the remaining *Nycticebus* species (6.04 mya, HPD: 8.39–4.02 mya). The divergence of the Bornean lineages from the *bengalensis* clade was estimated at 3.86 (5.41–2.55 HPD) mya. Among Bornean clades, the northern clade diverged at 1.93 (2.77–1.28 HPD) mya, and the western and eastern Borneo clades separated at 1.68 (2.38–1.09 HPD) mya. The MRCA of the northern Borneo clade was estimated at 0.3 (0.49-0.15 HPD) mya, the western Borneo clade at 0.73 (1.1–0.42 HPD) mya, and the eastern Borneo clade at 0.38 (0.8–0.1 HPD) mya. The split of the Java 1 clade from the *bengalensis* clade was estimated at 1.76 (2.67–1.07 HPD) mya, with the MRCA of the Java 1 clade estimated at 0.24 (0.39–0 HPD) mya. Within the *bengalensis* clade, the MRCA of the malayanus clade was estimated at 0.3 (0.51–0.14 HPD) mya, the Thai ‘dark’ Bengal clade at 0.41 (0.61–0.24 HPD) mya, and the northern Bengal clade at 0.41 (0.61–0.23 HPD) mya. The split between the northern and southern clades of *X. pygmaeus* was estimated at 1.18 (1.82–0.66 HPD) mya, with the MRCA of the northern pygmy clade estimated at 0.22 (0.4-0.11 HPD) mya and the MRCA of the southern pygmy clade at 0.35 (0.6–0.17 HPD) mya. The split between *N. coucang* (Sumatra) and the Java 2 clade was estimated at 1.08 (1.90–0.46 HPD) mya, with the MRCA of *N. coucang* (Sumatra) estimated at 0.33 (0.47–0.05 HPD) mya and the MRCA of the Java 2 clade at 0.34 (0.68–0.12 HPD) mya. For full HPD details of all estimated nodes, see Table 3 and Appendix A.

### 3.3. Craniometric Analysis

Based on the phylogenetic separation of *X. pygmaeus* into two lineages, we investigated whether cranial measurement differences in adult pygmy loris specimens supported such a division. For the full dataset of nine cranial measurements, an ungrouped morphometric analysis using PCA (N = 19 specimens without data gaps, including N = 7 northern lineage specimens and N = 12 southern lineage specimens; Appendix A) shows a distinction between the northern and southern pygmy loris lineages. While the northern lineage morphospace overlaps considerably with the southern lineage, the southern lineage occupies a notably broader morphospace (Figure 4 and Appendix A). The first principal component explains 63.6% of the variance, and the second principal component explains 12.0% (Figure 4). Cranial dimensions related to muzzle length are encompassed in the first principal component (greatest skull length, mandible length, and palate length) as well as bizygomatic breadth, while palate length and staphylion to basion length contributed most to the second principal component (Appendix A). Projections of specimen scores for the third and fourth principal components are provided in Appendix A, and underlying statistics for the PCA are provided in Appendix A.

We conducted pairwise comparisons for four of the cranial measurements related to muzzle length and shape and found that northern pygmy lorises had significantly smaller skull (greatest) length (SL mean = 49.14 mm, range = 44.08–50.85 mm, N = 3 males, 3 females; *t =* 2.3361, df = 19, *p* = 0.0306) and mandible length (ML mean = 30.76 mm, range = 27.04–33.29 mm, N = 3 males, 3 females; *t =* 2.79948, df = 20, *p* = 0.0112) than southern pygmy lorises (SL mean = 52.08 mm, range = 47.65–55.56 mm, N = 5 males, 4 females, 5 unknown sex; ML mean = 33.16 mm, range = 30.49–35.38 mm, N = 5 males, 4 females, 6 unknown sex; Figure 5, Appendix A). We found no significant difference in measurements of palate length (PL *t =* 0.5305, df = 18, *p* = 0.6022) or when specimens were grouped by sex for any of the four measurements (SL *t =* 1.2495, df = 15, *p* = 0.2306; ML *t =* 1.6059, df = 15, *p* = 0.1291; PL *t =* 0.2208, df = 14, *p* = 0.8284; PB *t =* 0.4857, df = 16, *p* = 0.6338). We found that northern pygmy lorises (PB mean = 17.18mm, range = 16.39–18.12mm, N = 3 males, 3 females) had smaller palate breadth than southern pygmy lorises (PB mean = 18.00mm, range = 16.16–19.69mm, N = 5 males, 5 females, 7 unknown sex), but this difference was not significant (*t =* 2.0736, df = 21, *p* = 0.0506; Figure 5, Appendix A).

## 4. Discussion

This research analyzed DNA sequences obtained from modern and historical museum specimens from multiple mitochondrial loci for all currently recognized species of slow and pygmy lorises to produce well-resolved phylogenetic relationships with predominantly strong support for major nodes (Figure 2). In general, the results agree with recent molecular studies [17,21,48,55,56,57,58], with some key differences that are likely due to the higher resolution of taxonomic and geographic sampling included in this study.

### 4.1. Pygmy Lorises: Ancient Divergence and Novel Diversity

Consistent with previous studies, we found strong support for the distinct and basal position of *X. pygmaeus* with respect to the slow lorises. Further, within *X. pygmaeus,* we found support for two reciprocally monophyletic lineages, one composed of samples from northern Vietnam/Laos/China and one composed of samples from southern Vietnam/Cambodia (Figure 1 and Figure 2). Our results confirm similar findings from a preliminary analysis on a smaller dataset (both of fewer samples and shorter sequence fragments; [59]). We also found significant differences in cranial measures between specimens from these ecographic groups with significant skull size and mandible reduction in more northern specimens (Figure 4 and Figure 5), consistent with previous studies [60].

We argue that the molecular divergence we present here, as well as evidence of ecographic morphological variation, confirms that the northern lineage represents a second taxon within the pygmy loris clade. Based on the type locality of the *X. pygmaeus* holotype NHMUK 1906.11.6.2 as described by Bonhote ([61], Nha Trang, Vietnam), *X. pygmaeus* refers to the southern taxon. For the northern taxon, the name *X. intermedius* (*Nycticebus intermedius* Dao Van Tien, 1960 [62]) is available, as the type locality for this taxon is Hòa Bình, and available *cytb* sequence data for this type specimen (HZM 069; [21]) clusters within the northern lineage (Appendix A, Appendix A). We discuss our results in the context of the taxonomic history of this taxon and propose a revised description for it below.

*N. intermedius* was first described by Dao Van Tien in 1960 [62], but subsequent genetic and morphological analyses of the holotype revealed it to be a synonym of what was then *N. pygmaeus* [17,63,64]. However, other researchers continued to argue for an additional pygmy loris taxon based on differences in pelage, body size, and anterior dentition, but with unclear evidence [25,65,66,67]. In particular, potential differences in pelage are challenging to diagnose due to the seasonal variation in pelage and body weight as exhibited by the pygmy loris [24].

Here, with unprecedented geographic coverage and sample size, we show that clear separation among mitochondrial lineages of pygmy loris coincides with significant differences in skull and mandible length. We also show that the northern lineage includes specimens that were historically identified as *N. intermedius*. While no skull is available for the type specimen HZM 069, we measured the skulls of two specimens identified as *N. intermedius* by their collectors (HZM 970, an adult male collected in Lục Yên, Yên Bái Province in Northeast Vietnam on 16 April 1964, and IEBRZM 56, an adult female collected in Na Hang, Tuyen Quang Province, in Northwest Vietnam on 22 October 1965; Appendix A). The skull (greater) lengths and mandible lengths of these *N. intermedius* specimens fall squarely within the range of values for the northern pygmy loris (50 mm or less) and are smaller than the average for the southern pygmy loris (Figure 4 and Figure 5, Appendix A).

Ravosa [60] discussed that the latitudinal differentiation within *X. pygmaeus* in skull size may be related to some ‘cline-dependent ecological factor’. Here, we provide confirmation of this result and show it coincides with molecular evidence, but the underlying cause for this variation and if it may relate to historical character displacement remains unclear. As we describe below, the climatic seasonality experienced by the northern lineage is starkly different from that experienced by the southern lineage (both historically and currently), and so further research in this area might be quite fruitful. For example, while both lineages seem to show some seasonal variation in pelage [25], further research is needed to determine whether this variation might be more extreme in the northern lineage. In addition, analyses of additional craniofacial measurements, soft tissue measurements, molar measurements, or postcranial measurements might provide useful diagnostic character information to further improve the taxonomic revision we present here.

A common zoogeographic barrier for vertebrates in the region is associated with the Hai Van Pass located between Quang Nam and Thua Thien Hue provinces, Vietnam (16.200 N, 108.125 E, 2846 m a.s.l.). Indeed, we have two samples of *Xanthonycticebus* from Quang Nam Province included in this study from different parts of the province; one sample clusters with the northern lineage, and one clusters with the southern lineage (Table 1). Both samples were collected during fieldwork for this research in Nam Giang District, and the sample that clusters with the southern lineage was collected in a more southerly location within the district (Chà Val). Importantly, the Hai Van Pass is to the north of both of these sample locations; thus, if a zoogeographic barrier was behind the divergence of these lineages, it might be a different one or the divergence may have been more related to known or historical climatic or ecotone differences on either side of the pass [68,69]. In addition, the distribution of the northern lineage in Laos remains unclear. Unfortunately, we are missing genetic information from any of the specimens collected from southern Laos on the Plateau Bolovens, which is at a slightly more southern latitude than Hai Van. Further research to delineate the current distributions of the lineages, including the extent of their range boundaries in southern Laos and whether they are in peripatry or sympatry in Quang Nam, will be important. Such work to document ecological preferences and conduct habitat suitability analysis may reveal whether the current distribution may be the result of secondary contact [70].

Importantly, the pairwise genetic distance (1.9%) and divergence estimate for the two pygmy loris lineages as estimated here (1.18, 1.82–0.66 HPD mya) is comparable to many currently recognized species in the genus *Nycticebus* (Figure 3, Table 2 and Table 3). We are confident these are strong estimates because our divergence date estimates for major nodes are within the confidence intervals for other recent analyses, including those that were estimated using nuclear markers and a combination of mitochondrial and nuclear markers. Our estimate for the *Nycticebus-Xanthonycticebus* MRCA was 11.34 mya (14.17–8.59 mya HPD) in the mid-Miocene, which overlaps the confidence interval studies that used a mitochondrial marker (10.9 mya; 7.6–14.5 mya; [21]), nuclear markers (3.5–10.1 mya; [55]) or a combination of both (18.4 mya; 10.2–26.9 mya; [57]).

Our divergence date estimates also lend support to the findings of Pozzi et al. [21] and Li et al. [56], showing a wide gap (roughly 6 million years) between the divergence of *Xanthonycticebus* and *Nycticebus* (6.04, 8.39–4.02 HPD mya) and the argument of Nekaris and Nijman [18] proposing a new genus name for pygmy lorises. The relatively deep divergence between *Xanthonycticebus* and the slow lorises may relate to other major differences between them, e.g., *Xanthonycticebus* is known to live sympatrically with other *Nycticebus* spp., to have hairless ears, to give birth to twins regularly, and to exhibit a multi-male, multi-female rather than a uni-male, uni-female social system [9,18,71]. Further, the divergence between *Xanthonycticebus* and the slow lorises is as old (mid-Miocene) as the divergence between other lorisid genera such as *Sciurocheirus* and *Otolemur* [21].

A recent study of pygmy loris whole genomes [50] revealed additional important distinctions in the functional genomics of *Xanthonycticebus* compared to other mammals and provided insights into its population demographic history that may explain the current sympatry of *Xanthonycticebus* spp. with *N. bengalensis*. However, Li et al. [56] did not include sufficient geographic coverage within the group to provide support for the split within *Xanthonycticebus* that we document here, emphasizing the crucial importance of studies like this one, which can inform critical discoveries despite shallow, limited genetic availability from degraded museum specimens due to comprehensive geographic and taxonomic coverage. Nevertheless, confirmation of our proposed taxonomic revision using genomic or at least nuclear DNA data would strengthen our findings.

#### Redescription of *X. intermedius* and *X. pygmaeus*

Order Primates Linnaeus, 1758

Suborder Strepsirhini É. Geoffroy Saint-Hilaire, 1812

Family Lorisidae Gray, 1821

Genus *Xanthonycticebus* Nekaris and Nijman, 2022

*Xanthonycticebus intermedius* (Dao Van Tien, 1960)

English Name: Northern pygmy loris

Holotype: The holotype and type locality were described in detail by Dao Van Tien [62]. The holotype is currently stored in the Hanoi Zoological Museum, Hanoi, Vietnam, under registration number 069. It is an adult female skin, collected from Hoà Bình, Vietnam (21.83 N, 105.33 E, 300 m a.s.l.) on 26 Jan 1957. Head and body length was measured at 230 mm (T 13, HF 45). The specimen was sequenced for the mitochondrial *cytb* gene for Pozzi et al. [21], GenBank Accession No. KP410626.

Additional specimens examined: (1) Adult male, skin and skull, FMNH 32499 (Figure 6) deposited in the Field Museum of Natural History, Chicago, IL. Collected on 3 March 1929 by R. W. Hendee during the Kelley–Roosevelts Expedition in Lai Chau, Lai Chau Province, Vietnam (22.06 N, 103.16 E, 907 m a.s.l.). Head and body length are measured at 248 mm (T 15.5, HF 53). Skull measurements are: greater skull length (SL), 50.7 mm; mandible length (ML), 33.29 mm; palate length (PL), 19.34 mm; and palate breadth (PB), 17.2 mm (Appendix A). (2) Adult male, skin and skull, HZM 970 deposited in the Hanoi Zoological Museum, Hanoi, Vietnam. Collected in Lục Yên, Yên Bái Province in Northeast Vietnam on 16 April 1964. Skull measurements are: SL, 48.95 mm; ML, 30.11 mm; PL, 19.45 mm; and PB, 17.39 mm (Appendix A). (3) Adult female, skin and skull, IEBRZM 56, deposited in the IEBR Zoological Museum, Hanoi, Vietnam. Collected in Na Hang, Tuyen Quang Province in Northwest Vietnam on 22 October 1965. Skull measurements are: SL, 49.81 mm; ML, 32.54 mm; PL, 19.89 mm; and PB, 17.47 mm (Appendix A).

Description and diagnosis: *X. intermedius* is distinctive in having a smaller skull length (consistently 50 mm or less) and mandible length (consistently 33 mm or less), resulting in a shorter muzzle (Figure 4, Figure 5, Figure 6 and Figure 7, Appendix A) compared to *X. pygmaeus*. Both *Xanthonycticebus* taxa exhibit finely textured pelage that is reddish buff with a medium to dark brown dorsal stripe and head forks [18,72] with seasonal variation in pelage, including crown coloration change and almost complete loss of the dorsal stripe [25], but *X. intermedius* often appears ‘fluffier’ with slightly longer body hair length in general (Figure 6 and Figure 7). The average head and body length of *X. intermedius* is 230 mm (range 214-248 mm; N = 4 specimens; Appendix A). Molecular estimates of the divergence between *X. pygmaeus* and *X. intermedius* are similar to or predate divergences between well-recognized species within other strepsirrhines (Figure 3).

Distribution: *X. intermedius* occurs in mainland Southeast Asia east of the Mekong River, but also historically in northern Laos PDR as far west as Phôngsali (21.59 N, 102.25 E) [18]. In Vietnam, it is found north of the Hai Van Pass in Quang Nam Province (16.200 N, 108.125 E, 2846 m a.s.l.) and also in southern China (historically north to Lüchun (23.00 N, 104.67 E) [18]. Further field surveys should be undertaken to determine whether *X. intermedius* may be sympatric with *X. pygmaeus* in parts of Quang Nam Province south of Hai Van and to determine the boundary of both species’ distributions in southern Laos, where there is some evidence of potential sympatry [65].

Order Primates Linnaeus, 1758

Suborder Strepsirhini É. Geoffroy Saint-Hilaire, 1812

Family Lorisidae Gray, 1821

Genus *Xanthonycticebus* Nekaris and Nijman, 2022

*Xanthonycticebus pygmaeus* (Bonhote, 1907)

English Name: Southern pygmy loris

Holotype: The holotype and type locality are described in detail by Bonhote [61]. The holotype is a juvenile male collected by J. Vassal on 13 November 1905 in Nha Trang, Vietnam (12.24 N, 109.19 E), deposited in the Natural History Museum London under registration number 1906.11.6.2.

Additional specimens examined: (1) Adult male, skin and skull, FMNH 46827 deposited in the Field Museum of Natural History, Chicago, IL. Collected on 24 March 1937 by W. H. Osgood in Buon Ma Thuot, Dak Lak Province, Vietnam (12.657 N, 108.03 E). Head and body length are measured at 265 mm (T 15, HF 54). Skull measurements are: greater skull length (SL), 55.56 mm; mandible length (ML), 34.76 mm; palate length (PL), 22.77 mm; and palate breadth (PB), 17.72 mm (Appendix A). (2) Adult female, skin, skull, and skeleton, FMNH 46825 deposited in the Field Museum of Natural History, Chicago, IL. Collected on 18 March 1937 by W. H. Osgood in Buon Ma Thuot, Dak Lak Province, Vietnam (12.657 N, 108.03 E). Head and body length are measured at 270 mm (T 15, HF 56). Skull measurements are: SL, 54.64 mm; ML, 33.64 mm; PL, 20.82 mm; and PB, 18.26 mm (Appendix A). This specimen was sequenced as a part of this study for the mitochondrial *CO1* gene (Appendix A, Appendix A), GenBank Accession No. OQ518107. (3) Adult male, skin, and skull, NMNH 258234 deposited in the National Museum of Natural History, Washington, D.C. Collected in Lam Dong, Bao Loc Township, Vietnam (11.573 N, 107.806 E, 800 m a.s.l.). Head and body length are measured at 235 mm (T 13, HF 50). Skull measurements are: SL, 50.82 mm; ML, 33.26 mm; PL, 18.32 mm; PB, 17.96 mm (Appendix A). This specimen was sequenced as a part of this study for the mitochondrial *CO1* gene (Appendix A, Appendix A), GenBank Accession No. OQ518103. (4) Adult male, skull, HZM 759 deposited in the Hanoi Zoological Museum, Hanoi, Vietnam. Collected in Ro Coi, Sa Thay, Kon Tum Province, Vietnam (14.51 N, 107.67 E) on 9 June 1980. Head and body length are measured at 260mm (T 10, HF 50). Skull measurements are: SL, 50.75 mm; ML, 31.88 mm; PL, 19.44 mm; and PB, 17.05 mm (Appendix A). (5) Adult skin and skull, IEBRZM 1879, deposited in the IEBR Zoological Museum, Hanoi, Vietnam. Collected in An Khê, Gia Lai Province, Vietnam (14.01 N, 108.71 E) on 19 April 1978 by Dinh Van C. Skull measurements are: SL, 52.09 mm; ML, 32.74 mm; PL, 20.02 mm; and PB, 18.51 mm (Appendix A).

Description and diagnosis: *X. pygmaeus* has a skull length of 55 mm or less and a mandible length of 35 mm or less (Appendix A). Pelage is slightly shorter in general than in *X. intermedius* (Figure 7). The average head and body length of *X. pygmaeus* is 264 mm (range 221–320 mm; N = 10 specimens; Appendix A).

Distribution: *X. pygmaeus* occurs in mainland Southeast Asia east of the Mekong River, in eastern Cambodia [72], in Vietnam as far north as the Hai Van Pass in Quang Nam Province (16.200 N, 108.125 E, 2846 m a.s.l.), and historically south to the vicinity of Ho Chi Minh City (10.75 N, 106.66 E) [18] and also southern Laos PDR [65]. Further field surveys should be undertaken to determine the boundary of *X. pygmaeus* distribution in southern Laos and the extent of potential sympatry with *X. intermedius* [65].

This published work and the taxonomic revisions it contains have been registered in ZooBank, the online registration system for the ICZN. The ZooBank LSIDs (Life Science Identifiers) can be resolved and the associated information viewed through any standard web browser by appending the LSID to the prefix “http://zoobank.org/”. The LSID for this publication is: urn:lsid:zoobank.org:pub: FE21B1AF-50F8-409F-9057-C6CB1B718C7B (accessed on 24 February 2023).

### 4.2. Support for Additional Diversity within Nycticebus

We present strong evidence for at least three distinct lineages of slow lorises within Borneo, namely a northern Borneo group that also includes a sample from Tawi Tawi Island in the Philippines, a western Borneo group, and an eastern Borneo group. Our phylogenetic tree also suggests additional structuring within the western Borneo group. Munds et al. [16] distinguished among four lineages within *N. menagensis* based on pelage characteristics. Our results present additional support for more than one lineage within Borneo, including potential support for the newly proposed taxon *N. kayan* as a separate lineage from *N. menagensis*. However, samples that we identified morphologically based on pelage characters following Munds et al. [16] as *N. menagensis, N. kayan, N. bancanus*, and *N. borneanus* are assorted among the three molecular lineages identified here. We suggest further systematic research on the slow lorises of Borneo to address potential challenges in specimen identification or to suggest description adjustments to clarify any discrepancies between molecular and morphological character diagnosis.

Our results also strongly suggest further research is needed to clarify the status of Sumatran/Malay slow lorises. Here, we show that *N. coucang* is paraphyletic, with *N. coucang* from peninsular Malaysia (mostly consisting of the putative subspecies *N. c. malayanus*) as distinct from Sumatran *N. coucang,* clustering within the *N. bengalensis* clade (Figure 2). The putative *N. c. tennaserimensis* also clusters within the *N. bengalensis* clade as a sister to the *malayanus* group. We also recover *N. hilleri* as a distinct lineage from Sumatran *N. coucang,* lending strong support to its distinction as the second Sumatran form. However, our analysis is lacking in samples with known provenance from southern Sumatra, so adding those would be key to providing further clarification on Sumatran diversity.

Additionally, within the *N. bengalensis* clade are groups of *N. bengalensis* samples denoting some additional genetic variation, as has been reported in previous studies [51], but unlike the Bornean groups, these do not consistently correspond to geographic areas. One group is entirely comprised of samples of the ‘dark morph’ *N. bengalensis* from Thailand, but the ‘northern’ *N. bengalensis* group includes samples from as far apart from one another as Assam, northern Burma/Myanmar, northern Vietnam, and Phu Quoc Island off the southern coast of Vietnam. The *N. bengalensis* group is less strongly supported than other lineages as a monophyletic group when the sample from Langkawi Island is included in the group, but the sample is likely within this group or sister to it [32]. The divergence date estimates and mean pairwise genetic distances for the subclade splits within this group appear quite recent (Table 2 and Appendix A), and given that some clades do not have very strong node support, further taxonomic splitting might not be appropriate within *N. bengalensis*, but the presence of a high degree of potentially unrecognized diversity is obvious. It will be helpful to include additional samples from this group with known origins, perhaps especially from northwestern Thailand, peninsular Thailand, and Myanmar, in a future analysis that also includes nuclear DNA.

### 4.3. Placement of N. javanicus

Notably, the placement of *N. javanicus* differed in this study from previous research. In previous studies using single-locus mitochondrial datasets, *N. javanicus* and *N. bengalensis* are recovered as sister taxa [17,21]. However, in Chen et al. [73], the placement of *N. javanicus* was less clear as only one *N. javanicus* sample was included, and it was of unknown provenance. The four *cytb* sequences representing *N. javanicus* in the studies by Pozzi [21] and Roos [17] were similarly of unclear provenance. In this study, we included two samples used by Pozzi and Roos’ studies (NJAV and N3), and they did recover as sisters to *N. bengalensis*, but the rest of the *N. javanicus* specimens included in this study, all novel sequences with known provenance from museum collections, were strongly supported as sisters to the Sumatran *N. coucang* samples (Figure 2, Table 1).

This very different placement of the phylogenetic relationship of *N. javanicus* to the rest of the genus could be explained by greater taxonomic and geographic sampling in this study as compared to previous studies. Our results could lend support to additional diversity within *N. javanicus,* as has been proposed through morphological analysis [20]. Additional samples with known provenance from Java and the analysis of additional loci, especially nuclear loci, would help to clarify the various plausible explanations for our findings here and described above. The lack of nuclear loci in our analysis is likely the reason why some of the deeper nodes in the tree lack strong support [57]. The inclusion of nuclear loci in a future analysis would help to clarify the placement and taxonomic status of *N. javanicus* and also weigh the potential importance of incomplete lineage sorting on phylogenetic inference throughout *Nycticebus* due to the rapid divergence of the rest of the genus after the split from *Xanthonycticebus.*

### 4.4. Historical Wildlife Trade and Issues of Known Provenance

Further analysis could also illuminate potentially pervasive issues of historical wildlife trade and other issues of known provenance for phylogenetic inference based on datasets derived from museum specimens. In this study, a museum specimen collected in the 1930s (AMNH 102027) was included that had a tagged origin from Singapore, but it clustered within the western Borneo group in our analysis (Figure 1), presenting potential evidence of historical wildlife trade. Indeed, 100 years ago, Singapore as well as Johor Lama were major trading centers for all kinds of primates and other wildlife [74,75,76]. Thus, it is within the realm of possibility that for AMNH 102027 or other specimens, such as NMNH 114151 from Johor Lama, the tagged provenance is incorrect or may be the result of trade. When records and collector’s notes are available to cross-check provenance information, these clues can be vital to determine whether the specimen may have originated in trade or could represent historical diversity that is no longer observed today because of loss of habitat.

Other *N. coucang* and *N. javanicus* specimens without a clear provenance (NMNH 14290) or that we know originated in trade (N3, NJAVA) clustered at the base of the *N. bengalensis* clade, disparate from other members of their taxa included in the analysis. Again, while these results could represent true, additional diversity in what is already a highly diverse genus, it is also possible that these specimens were collected from wildlife markets or traders or could even represent anthropogenic hybrids in more recently collected samples. Many previous studies applying molecular genetics to the inference of slow loris phylogeny have relied on samples collected from trade and, perhaps as a result, have conflicted with each other on the issue of the presence of hybrids. For example, Chen et al. [73] did not find reciprocal monophyly between *N. coucang* and *N. bengalensis* and suggested that their results, based on a very small sample size, may be evidence for introgression between both species in peninsular Thailand, as proposed by Groves [72]. However, other studies using larger sample sizes (both in terms of sequence length and taxonomic sampling) did not find evidence of introgression [17,21].

As discussed above, more concerted sampling efforts and also interdisciplinary research to account for both historical and current wildlife trade flows [77] will be important to illuminate the extent of hybrid individuals represented in the sample in this study and in both museum specimen collections and the GenBank repository broadly due to origins from trade or zoos (or the use of outdated taxonomy, as is the case with AJ309867, which is listed as *N. coucang* but clearly clusters with *N. bengalensis* in our analysis, Table 1, Figure 2). Such efforts will greatly improve our understanding of the nature of slow loris diversity and that of other highly traded wildlife [10]. Furthermore, as genomic analyses of historical museum collections continue to expand rapidly, processes to update vouchered information on provenance to account for historical wildlife flows are sorely needed [14].

### 4.5. Informing Conservation Management

Assessments of genetic relationships within the slow lorises have great potential to improve conservation efforts for these globally threatened primates. The most recent IUCN assessment in 2020 recognizes nine pygmy and slow loris species, including *N. hilleri*, with all except three recently upgraded to either endangered or critically endangered due to increasing and persistent threats from overexploitation for illicit wildlife trade at local and global scales [78]. The results we present here, especially by providing a great deal more vouchered genetic information across the distributions of these species, will help to confirm population limits for accurate assessments and to provide reference information to identify the provenance of confiscations, which can both help to identify patterns and drivers of wildlife trade and can support more successful releases of healthy individuals. For example, our results supporting two distinct lineages within the pygmy loris means that confiscated animals from the southern lineage found in the north should be released back to the south for the highest likelihood of survival. This information is already being used by species management partners in Vietnam including the major rescue centers, and the genetic ID of animals when they arrive is integrated into rescue and release protocols [79]. In addition, the results have implications for the species management and captive breeding programs of global captive populations of pygmy lorises.

As discussed above, both museum collections and the GenBank repository are essential sources of vouchered information to enable the identification of species and provenance of wildlife-trade confiscated specimens at the global scales at which trade is occurring, which could serve to identify patterns and drivers of trade in an interdisciplinary systems framework [14]. However, the utility of these resources is seriously hampered if the information in these repositories is of unknown or incorrect provenance. Large taxonomic and geographic gaps in coverage also impede the potential of these resources to inform wildlife trade research and management. For example, our analysis is still lacking samples from large swaths of the *N. bengalensis* range, as well as peninsular Thailand, southern Sumatra, and southern Borneo, which remains an issue for many rescue centers within their distribution to diagnose appropriate provenance for the release of healthy individuals back to the wild [51,80,81].

Lastly, while methods to collect genomic data from museum specimens are available, these require quality standards that less than a fourth of our extracted samples met in terms of genomic DNA concentration and quality. Thus, to achieve the broadest and most complete sample coverage in terms of what was available with current technologies, we used Sanger sequencing of mtDNA. However, our results put forward key advances to lay the groundwork for a vouchered reference database and illuminates pervasive issues to be resolved in order to enable high throughput studies in the future that will make possible the diagnosis of wildlife-trade-confiscated specimens at scale.

## Figures and Tables

**Figure 1 genes-14-00643-f001:**
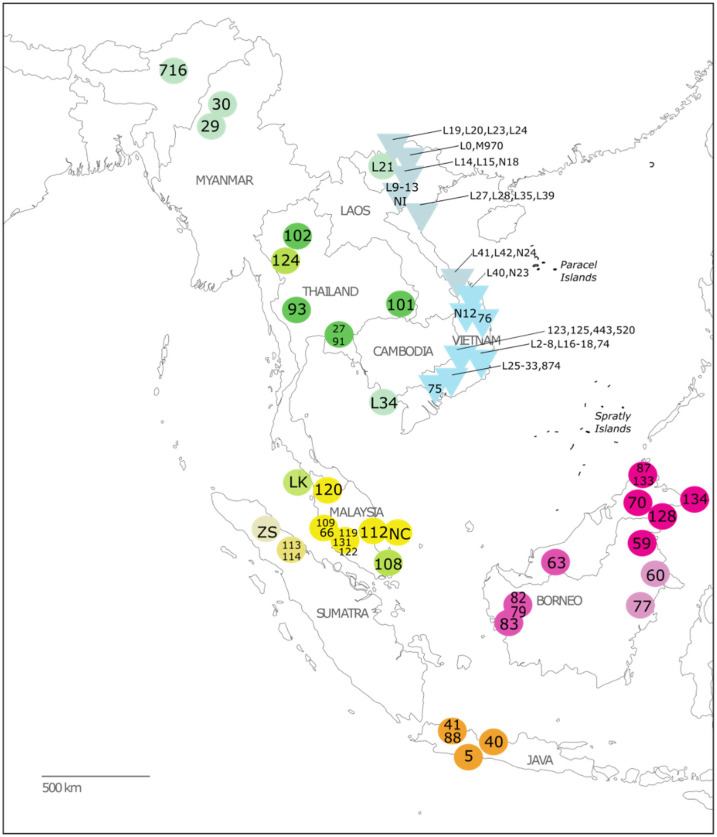
Geographic coverage of the samples included in this study for which provenance is known. Triangles represent pygmy loris samples, and circles represent slow loris samples. Colors represent clades, as shown in Figure 2. Sample IDs correspond to details provided in Table 1 and Appendix A.

**Figure 2 genes-14-00643-f002:**
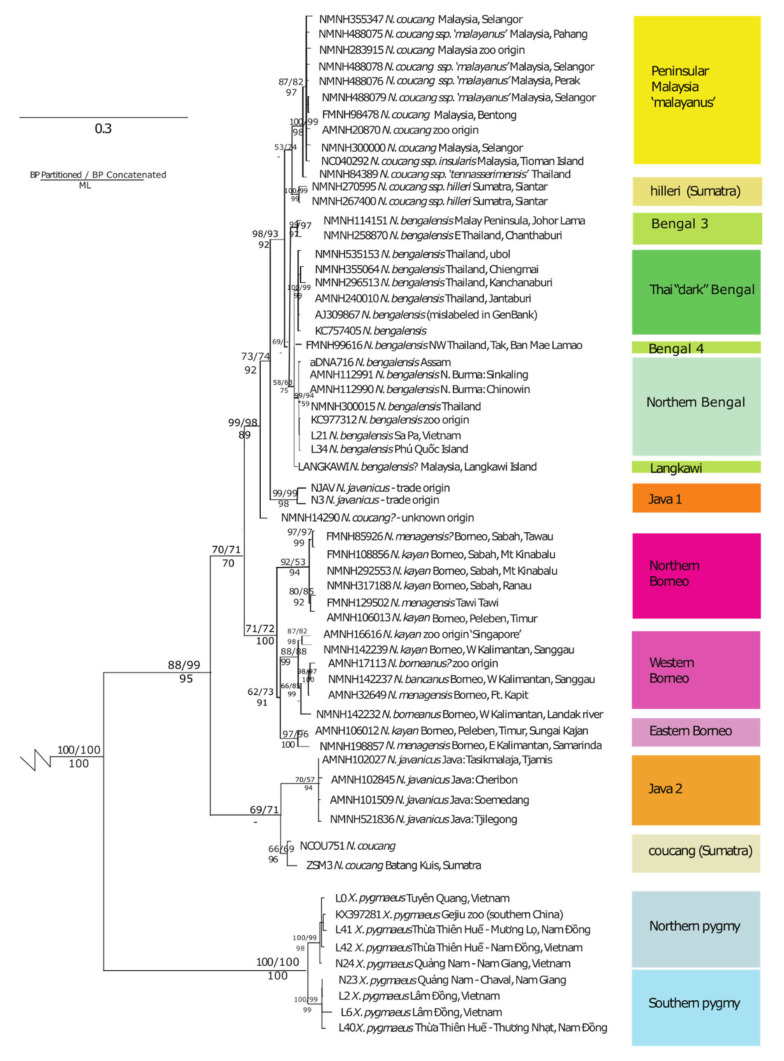
Phylogenetic tree inferred from maximum likelihood and Bayesian analyses. Node values are reported for Bayesian posterior probability above the line (both partitioned and concatenated analyses results are shown; 100 means PP = 1.0) and maximum likelihood bootstrap support (%) below the line. Scale bar represents substitutions per nucleotide.

**Figure 3 genes-14-00643-f003:**
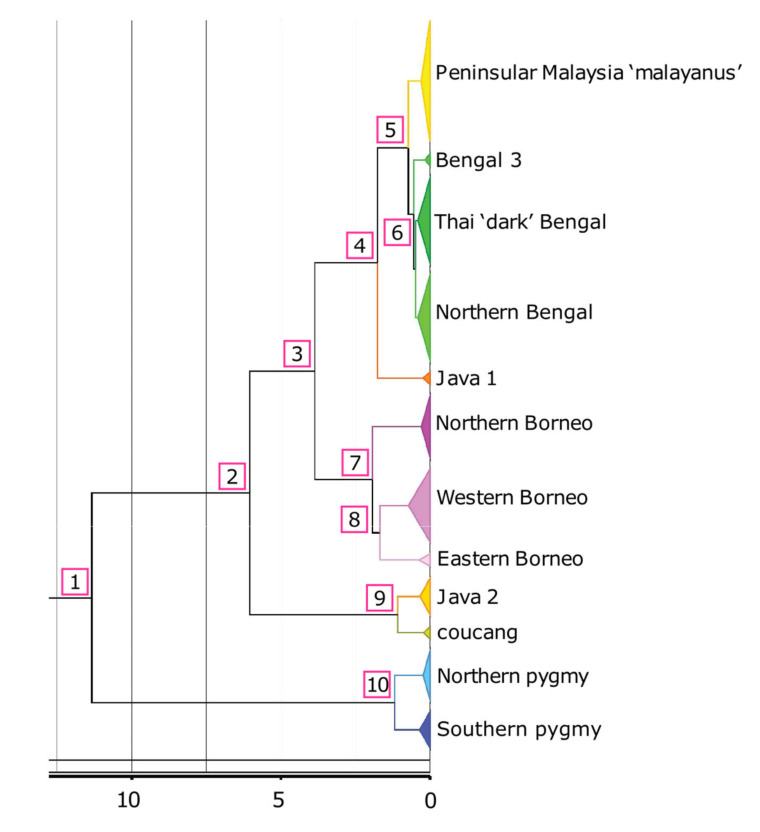
Estimated divergence ages of clades corresponding to those shown in Figure 2. Specific estimates with 95% highest probability densities (HPDs) are provided in Table 3. A time scale is given below the tree (values are millions of years ago).

**Figure 4 genes-14-00643-f004:**
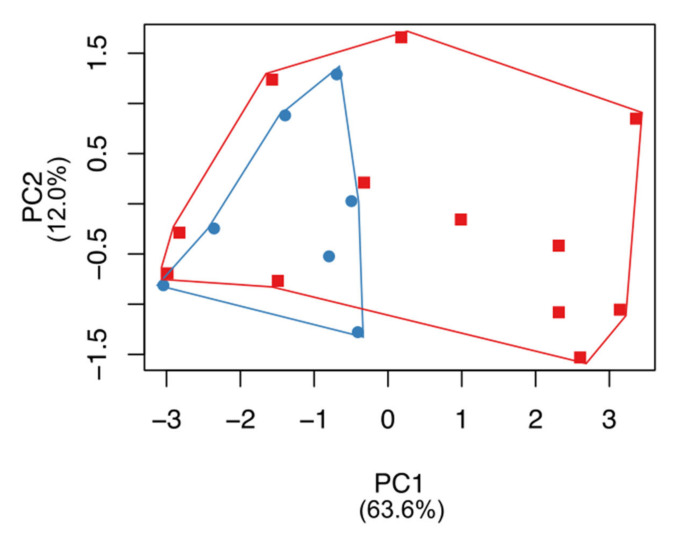
Morphometric comparison (principal component analysis performed on nine cranial measurements) among pygmy loris specimens representing northern (blue circle) and southern (red square) lineages. A projection of specimen scores for the first and second principal components is shown, with variance explained by each component (the third and fourth principal components are shown in Appendix A, and underlying statistics are provided in Appendix A).

**Figure 5 genes-14-00643-f005:**
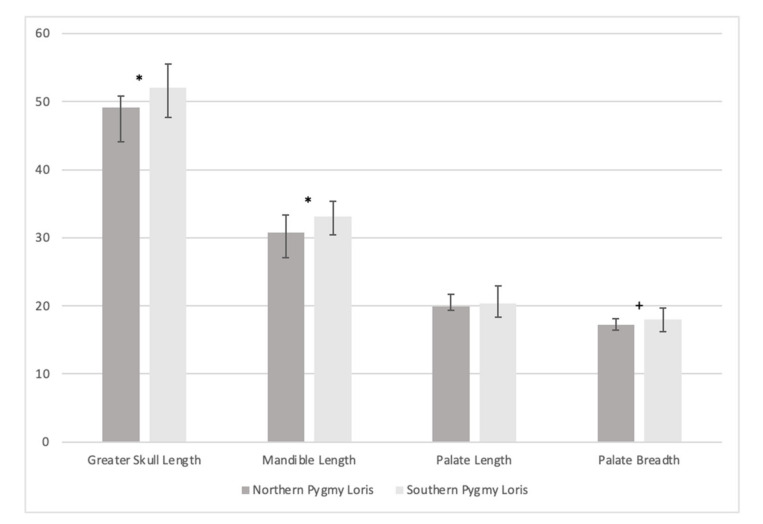
Plot of mean craniometric distances (mm) of adult northern and southern pygmy loris specimens across four measures, showing a significant difference (*) between northern and southern specimens for greater skull length (*t =* 2.3361, df = 19, *p* = 0.0306), mandible length (*t =* 2.79948, df = 20, *p* = 0.0112), and a near significant difference (+) for palate breadth (*t =* 2.0736, df = 21, *p* = 0.0506). Error bars represent the minimum and maximum distances measured.

**Figure 6 genes-14-00643-f006:**
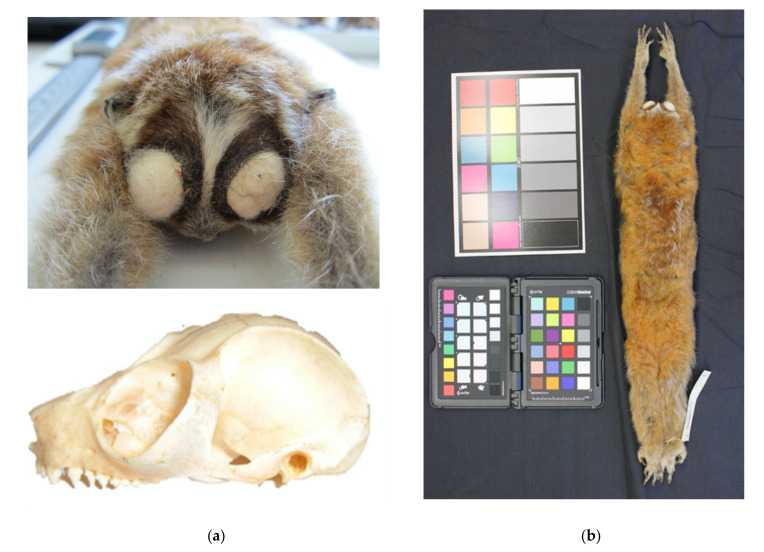
*Xanthonycticebus intermedius* specimen FMNH 32499: (**a**) Above: Facial mask (Photo by R. Munds), Below: Skull (Photo by K.A.I. Nekaris); (**b**) Full Dorsal View (Photo by M. Blair).

**Figure 7 genes-14-00643-f007:**
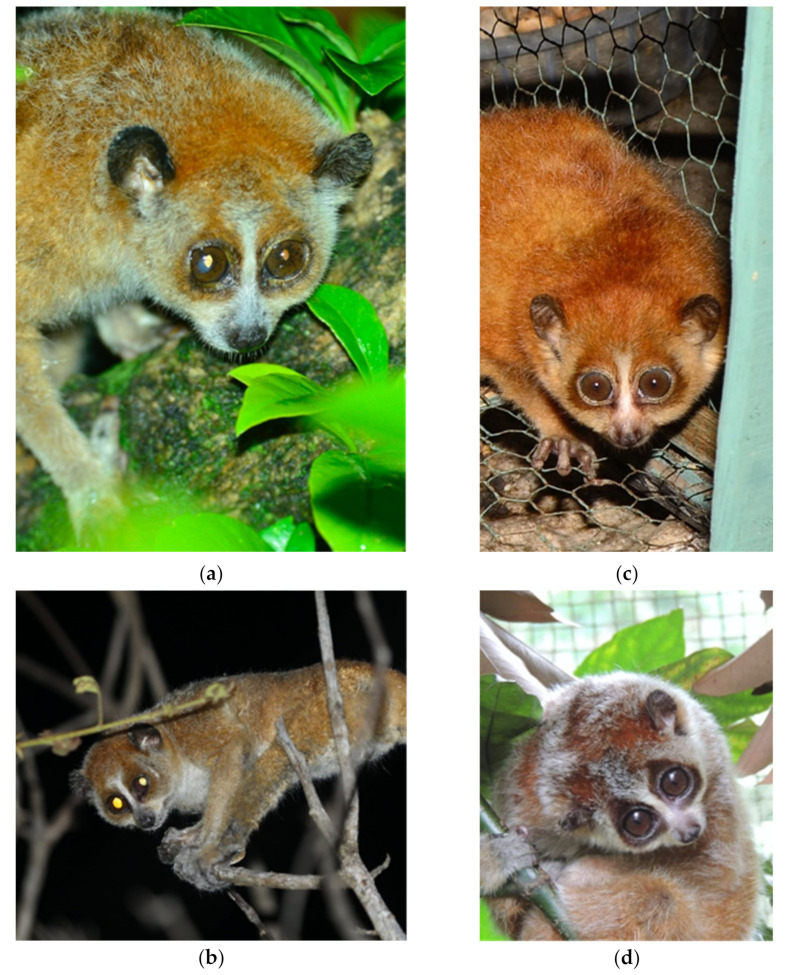
Photos depicting the longer muzzle in the southern pygmy loris: (**a**) Lam Dong Province, Vietnam in 2015, Photo by H. Thach; (**b**) Bu Gia Map National Park, Binh Phuoc Province, Vietnam in 2014, Photo by Kieu Dinh Tháp); and a shorter muzzle in the northern pygmy loris: (**c**) Tuyen Quang Province, Vietnam in 2013; Photo by H. Thach; (**d**) EPRC, Cuc Phuong National Park, Vietnam in 2014, Courtesy of T. Nadler, Photo by M. Blair) also showing the full range seasonal pelage change in the pygmy loris: (**c**) summer coloring; (**d**) winter coloring.

**Table 2 genes-14-00643-t002:** Mean pairwise genetic distances within and among major clades based on the combined mitochondrial dataset and following the clade numbers in Table 1.

	**1**	**2**	**3**	**4**	**5**	**6**	**7**	**8**	**9**	**n**
1	**0.0056**	0.0132	0.05	0.0483	0.043	0.0537	0.0372	0.1218	0.1256	0.0218
2		**0.0009**	-	-	-	-	0.0772	0.0936	0.1176	-
3			**0.0038**	0.033	0.0304	0.0653	-	0.1216	0.125	0.0487
4				**0.0065**	0.0246	0.0661	-	0.1192	0.1231	0.0568
5					**0.0054**	0.062	-	0.1121	0.0955	0.0712
6						**0.0026**	-	0.1121	0.1126	0.0873
7							**0.0044**	0.0987	0.1231	-
8								**0.0069**	0.0187	0.16
9									**0.0057**	0.1209
n										**-**

- denotes values that were not calculable.

**Table 3 genes-14-00643-t003:** Estimated divergence ages with 95% highest probability densities (HPDs) of nodes as presented in the phylogeny (in mya, million years ago). Node numbers are defined in Figure 3.

Node	Age (mya)	HPD	Node	Age (mya)	HPD
1	11.34	8.59–14.17	Bengal 3 MRCA	0.17	0.05–0.33
2	6.04	4.02–8.39	‘Dark’ Bengal MRCA	0.41	0.24–0.61
3	3.86	2.55–5.41	Northern Bengal MRCA	0.41	0.23–0.61
4	1.76	1.07–2.67	Java 1 MRCA	0.24	0–0.39
5	0.73	0.46–1.06	Northern Borneo MRCA	0.3	0.15–0.49
6	0.55	0.35–0.81	Western Borneo MRCA	0.73	0.42–1.10
7	1.93	1.28–2.77	Eastern Borneo MRCA	0.38	0.10–0.80
8	1.68	1.09–2.38	Java 2 MRCA	0.34	0.12–0.68
9	1.08	0.46–1.90	*N. coucang* (Sumatra) MRCA	0.22	0.05–0.47
10	1.18	0.66–1.82	Northern pygmy MRCA	0.22	0.11–0.40
‘*malayanus*’ MRCA *	0.30	0.14–0.51	Southern pygmy MRCA	0.35	0.17–0.60

* MRCA is the most recent common ancestor.

## Data Availability

DNA sequence data generated for this study have been deposited to GenBank under the following accession numbers: OQ518052-OQ518145, OQ555473-OQ555600.

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
