# Peer review of "Molecular Phylogenetic Relationships and Unveiling Novel Genetic Diversity among Slow and Pygmy Lorises, including Resurrection of Xanthonycticebus intermedius"

_genes, 2023, doi:10.3390/genes14030643_

Round 1

Reviewer 1 Report

This is a well written manuscript that significantly increases our understanding of slow and pygmy loris phylogenetics.  The documentation of two lineages of pygmy lorises is of particular interest with both phylogenetic and conservation implications.  You also do an excellent job of identifying where additional sampling is needed to offer further clarifications of slow loris taxonomy.

Typos that need to be correct:

opisthocranium is misspelled on lines 129-130 (opisthokranium is incorrect)

"In This" on line 779 should be "This" (delete In)

Author Response

Dear Reviewer 1:

Thank you for catching these typos, we fixed the spelling of "opisthocranion" on Line 129-130 and deleted 'In" on line 779 (now line 781).

Best regards,
Mary Blair on behalf of all co-authors.

Author Response

Dear Reviewer 2:

  1. We thank the reviewer very much for pointing out these important corrections. The information we provided was as written on specimen labels and in the museum collection databases. We have made the suggestions as corrected to Figure 1, Table 1, and Table S1, and will pass along these corrections to the respective museum collections so that their databases can also be updated.
  2. We thank the reviewer for pointing out the missing specimen information and location clarifications. We have corrected Figure 1 and Table 1 to include the missing specimen, and updated our discussion to include the suggested information.
  3. We have added pairwise genetic distance results to the text, specifically in the results section of the manuscript and we also added three new tables (two supplementary) with complete pairwise genetic distance information for all sequences, and summarized within and between clades.
  4. We thank the reviewer very much for their thoughtful suggestions and interesting ideas. Indeed in our discussion we propose as potential future studies the very followup analyses that could confirm these ideas, and we would very much welcome any collaborations.

Best regards,
Mary Blair on behalf of all co-authors.

Reviewer 3 Report

This paper is interesting, and the report of results and discussion are well done. I suggest a few minor considerations, and a major one, that is, the need to add a detailed multivariate morphometric analisys. I’m not an expert on the taxonomy of lorises, thus, I would suggest that, before reconsidering it for publication, this ms. should be evaluated by an expert taxonomist.

1.      Introduction

Line 97 and following: before concluding the Introduction, please add a short clarification on the fact that mtDNA is in practice a single linked gene that exclusively expresses maternal phylogenies. The results of this study should therefore be considered as a preliminary working hypothesis pending phylogenetic analyses of chromosomal DNA sequences.

2.      Materials and Methods 2.1. Sample collection

Line 111 Following approved protocols for destructive sampling as established …: add references, if available.

Line 122: For each museum specimen, photographs and measurements …: it would be useful to the reader who does not have the cited publications, add a table (possibly in the Supplementary) with a detailed list of the measured craniometric characters. It would also be appropriate to use these craniometric traits to conduct a multivariate analysis, e.g., principal component analysis or discriminant analysis, to be included in the results and check for congruence or discrepancy with the results of the molecular analyses.

Author Response

Dear Reviewer 3:

  1. We agree with the reviewer, and had already included this important information in our discussion section. We added some additional language to the end of the Introduction in response to this comment.
  2. We included a link to the AMNH guidelines in the revision.

  3. We thank the reviewer very much for this suggestion. We performed a PCA analysis as suggested and added definitions of the measurements to a new Table S3, which also includes the full PCA results. Addition of the PCA analysis improved the strength of our findings as it is highly appropriate for this dataset. We added two new figures, Figure 4 and S6, with the results and updated the text in the methods, results, and discussion sections to describe this new analysis and its implications for our argument. In the process of doing this new analysis we also discovered we had data for a few more specimens that could be included from the MNHN in Paris, and we added the details of those specimens into a revised Table S1. Also in response to this comment, we removed the definitions of the cranial measurements that had been in the text of the methods, and moved them to the new Table S3 (since now we included all nine measurements in the PCA analysis and not just four).

Thank you for your comments and suggestions, they greatly improved the manuscript.

Mary E. Blair on behalf of all co-authors